# Secretome Analysis of *Clavibacter nebraskensis* Strains Treated with Natural Xylem Sap In Vitro Predicts Involvement of Glycosyl Hydrolases and Proteases in Bacterial Aggressiveness

**DOI:** 10.3390/proteomes9010001

**Published:** 2021-01-09

**Authors:** Atta Soliman, Christof Rampitsch, James T. Tambong, Fouad Daayf

**Affiliations:** 1Department of Plant Science, University of Manitoba, Winnipeg, MB R3T 2N2, Canada; atta.soliman@canada.ca; 2Department of Genetics, Faculty of Agriculture, University of Tanta, Tanta, Gharbiya 31111, Egypt; 3Lethbridge Research and Development Centre, Agriculture and Agri-Food Canada, Lethbridge, AB T1J 4B1, Canada; 4Morden Research and Development Centre, Agriculture and Agri-Food Canada, Morden, MB R6M 1Y5, Canada; chris.rampitsch@canada.ca; 5Agriculture and Agri-Food Canada, Ottawa, ON K1A 0C6, Canada; james.tambong@canada.ca

**Keywords:** secretome, *Clavibacter*, xylem sap, corn, mass spectrometry

## Abstract

The Gram-positive bacterium *Clavibacter nebraskensis* (Cn) causes Goss’s wilt and leaf blight on corn in the North American Central Plains with yield losses as high as 30%. Cn strains vary in aggressiveness on corn, with highly aggressive strains causing much more serious symptoms and damage to crops. Since Cn inhabits the host xylem, we investigated differences in the secreted proteomes of Cn strains to determine whether these could account for phenotypic differences in aggressiveness. Highly and a weakly aggressive Cn strains (Cn14-15-1 and DOAB232, respectively) were cultured, in vitro, in the xylem sap of corn (CXS; host) and tomato (TXS; non-host). The secretome of the Cn strains were extracted and processed, and a comparative bottom-up proteomics approach with liquid chromatography–tandem mass spectrometry (LC–MS/MS) was used to determine their identities and concentration. Relative quantitation of peptides was based on precursor ion intensities to measure protein abundances. In total, 745 proteins were identified in xylem sap media. In CXS, a total of 658 and 396 proteins were identified in strains Cn14-5-1 and DOAB232, respectively. The unique and the differentially abundant proteins in the secretome of strain Cn14-5-1 were higher in either sap medium compared to DOAB232. These proteins were sorted using BLAST2GO and assigned to 12 cellular functional processes. Virulence factors, e.g., cellulase, β-glucosidase, β-galactosidase, chitinase, β-1,4-xylanase, and proteases were generally higher in abundance in the aggressive Cn isolate. This was corroborated by enzymatic activity assays of cellulase and protease in CXS. These proteins were either not detected or detected at significantly lower abundance levels in Cn strains grown in non-host xylem sap (tomato), suggesting potential factors involved in Cn–host (corn) interactions.

## 1. Introduction

The phytopathogenic bacterial genus *Clavibacter* belongs to the Microbacteriaceae family (Class *Actinobacteria*). Members of this genus are pathogens of economically important agricultural crops [1,2], causing significant yield losses [1,3]. One of the species, *Clavibacter nebraskensis*, is the causal agent of the Goss’s wilt and leaf blight disease of corn [4,5,6]. This bacterial pathogen causes foliar infections by entering wounds in leaf tissues produced by mechanical damage, e.g., hail, blowing sandy soil, farm machinery, and/or herbivore grazing [7,8,9]. After a heavy rain or hail, bacteria in infected debris splash onto wounded leaves allowing Cn to enter plant tissues. Disease symptoms first appear in the form of gray lesions with streaks of small water-soaked spots (freckles), before turning necrotic. Bacterial exudates appear shiny after drying on the infected tissues [10]. In systemic infection phase of the disease, the pathogen colonizes the xylem vessels and spreads throughout the plant [11], which might lead to colonization of the seeds, providing another source of inoculum [5]. Infection in the absence of wounding can also occur via natural openings under high humidity conditions [12]. Cn overwinters in corn stubble and in alternate host plants such as barnyardgrass (*Echinochloa crus-galli*), foxtail (*Setaria spp.*), shattercane (*Sorghum bicolor*), annual ryegrass, Johnson grass (*Sorghum halepense*), and large crabgrass [5,7,13,14].

A few studies have investigated *Clavibacter*–host interactions at the proteome level [15,16]. The secretome of *Clavibacter michiganensis* subsp. *michiganensis* (Cmm)*,* the causal agent of tomato stem canker, was investigated in minimal and xylem-mimicking media, revealing that Cmm responds to xylem sap similarly as to sugar-depleted media by secreting virulence factors such as Pat-1 and CelA [16]. In another study, Savidor and colleagues [15] investigated the secretome and showed that Cmm secretes multiple hydrolytic proteins, such as serine proteases and glycosyl hydrolases, which promote infection and disable plant defense components. Infected tomato responds by activating defense genes such as pathogenesis-related (PR) and other defense-related proteins [15]. These studies demonstrated the reliability of using xylem sap to better understand host–pathogen interactions, especially for Cmm. This xylem-sap approach could be a suitable medium for studying Cn–corn interactions in vitro. However, to date, no published reports exit on the use of xylem sap to study Cn.

Here, we report differences in the secreted proteomes between a highly aggressive (Cn14-5-1) and a weakly aggressive (DOAB232 = NCPPB2581 = LMG 5627^T^) strains of Cn grown in corn xylem sap (CXS). These differences were less marked or, even, not observed when the strains were grown in a non-host tomato xylem sap (TXS), indicating that the host sap might contain relevant factors involved in triggering or inducing Cn pathogenicity traits. Our results indicated that the Cn-secreted proteins might be involved in scavenging and mobilizing host plant nutrients for bacterial growth and proliferation. In addition, Cn deployed known virulence factors that affect the integrity of the host plant’s cell walls and membranes, e.g., cell wall-degrading enzymes (CWDEs) and proteases, especially in the aggressive isolate. The proteomics-based observations were supported by in vitro enzymatic assays of total cellulases and proteases, which showed significantly higher activities in the aggressive isolate, Cn14-5-1. Finally, secretion of chaperone proteins, part of a repair mechanism for stress-damaged proteins is potential evidence of the ability of Cn to counteract and evade the host’s defense strategies.

## 2. Materials and Methods

### 2.1. Bacterial Growth Conditions

Two Cn strains, the highly aggressive Cn14-5-1, and the weakly aggressive DOAB232^T^ (=NCPPB2581^T^), were streaked on nutrient broth yeast extract (NBY), containing 0.8% (*w*/*v*) nutrient broth, 0.2% (*w*/*v*) yeast extract, 0.2% (*w*/*v*) K_2_HPO_4_, 0.05% (*w*/*v*) KH_2_PO_4_, 0.5% (*w*/*v*) glucose, and 1.5% (*w*/*v*) agar. After autoclaving, 1 mL of 1 M MgSO_4_·7H_2_O was added [17]. The bacterial cultures were incubated at 23–24 °C for 4–5 days. The term aggressiveness of Cn strains refers to the level of leaf damage induced by tested strains.

### 2.2. Aggressiveness Assay

Corn plants were inoculated with the two Cn strains (Cn14-5-1 and DOAB232). Corn hybrid A4631G2 (RIB Pride Seeds), rated as Goss’s wilt-tolerant (2015 Crop Production Services Seed guide, Loveland, CO, USA), was used to rate the ability of the two strains to induce leaf damage. In brief, Cn strains were grown on NBY plates for 4–5 days at 23–24 °C, and then loops of bacterial colonies were suspended and diluted in phosphate buffer (10 mM potassium phosphate dibasic, and 10 mM potassium phosphate monobasic, pH 6.7) and the concentration was adjusted to 7 × 10^8^ cfu·mL^−1^. Three leaves (third, fourth, and fifth) of corn plants at the V4-V5 stage were wounded on both sides of the middle vein using a 1 mL disposable syringe plunger mounted with a 5 mm sand paper disk. Twenty microliters of inoculum were applied onto each wound. The wounded control plants were inoculated with phosphate buffer. Plants were placed in a humidity tent with 100% relative humidity overnight. The area under the disease progress curve (AUDPC) was calculated for both lesion length and disease severity as previously described [18]. The experiment was performed in 3 biological replicates and the entire experiment was repeated once.

### 2.3. Hydrogen Peroxide and Superoxide Localization

Hydrogen peroxide (H_2_O_2_) was visualized in the infected corn leaves using 3,3′-diaminobenzidine tetrahydrochloride (DAB; Sigma-Aldrich, St. Louis, MO, USA). Infected leaves were submerged in DAB solution (1 mg·mL^−1^ DAB in 50 mM Tris–acetate buffer (pH 5.0) and vacuumed for 5 min, and thereafter kept in the dark for 24 h. Infiltrated leaves were subsequently bleached using ethanol/glycerol/glacial acidic acid (3:1:1) for 15 min in a water bath at 95 °C. The latter step was repeated 3–4 times to clear out the remaining chlorophyll. Superoxide (O_2_^−^) was visualized using 0.5 mg·mL^−1^ nitroblue tetrazolium (NBT) (Sigma-Aldrich, St. Louis, MO, USA) dissolved in 10 mM potassium phosphate buffer (pH 7.6) for 6 h in the dark and then chlorophyll was bleached out as described above and photographed [19].

### 2.4. Xylem Sap Collection

Corn xylem sap was collected from 6–8-week-old plants grown in a greenhouse (22 °C, 18 h light and 16 °C, 8 h dark). Stems were cut 10–15 cm above the soil and cleaned twice using autoclaved wipes to remove leaking sap. The sap was collected using a micropipette for 1 h, filter-sterilized using 0.22 µm filters (Millipore, Burlington, MA, USA), and stored at −20 °C. Tomato xylem sap was collected from 4-week-old plants grown under the same conditions and using the same procedure.

### 2.5. Bacterial Induction with Xylem Sap

The two Cn strains were grown in NBY liquid media for 12 h at 23–24 °C. The concentration of the bacterial cultures was adjusted to OD_600_ = 1.0, and then equal volumes of the cultures were centrifuged for 20 min at 6000× *g*. The bacterial pellets were washed twice with sterilized distilled water. Bacterial pellets were suspended and incubated on an orbital shaker at 200 rpm for 12 h at room temperature in 20 mL in either corn xylem sap (CXS) or tomato xylem sap (TXS). The experiment was carried out with 3 biological replicates and the entire experiment was repeated once.

### 2.6. Protein Extraction, Reduction, Alkylation and Digestion

Bacterial cells were removed from the xylem sap by centrifugation for 20 min at 12,000× *g*, and the supernatant was filtered through 0.22 μm filters to remove any remaining cells. The filtrates were concentrated to approximately 300 µL using centrifugal devices (10 kDa cut-off, Millipore Sigma, Burlington, MA, USA). Proteins were harvested by precipitation in 10% (*w*/*v*) trichloroacetic acid (TCA)-acetone and 0.07% (*w*/*v*) dithiothreitol (DTT) overnight at −20 °C. Samples were then centrifuged at 12,000× *g* for 20 min and protein pellets washed 3 times with acetone–DTT. The final pellet was air-dried and suspended in 100 mM ammonium bicarbonate.

Proteins were reduced in 5 mM DTT and incubated for 25 min at 56 °C. After cooling to room temperature, alkylation was performed in 14 mM iodoacetamide in the dark for 30 min. Protein samples were then dialyzed using Slide-A-Lyzer MINI Dialysis Device (ThermoFisher Scientific, Waltham, MA, USA) in 100 mM ammonium bicarbonate buffer for 2 h at room temperature, and then overnight in fresh dialysis buffer at 4 °C. Proteins were quantified using a Bradford assay [20] with bovine serum albumin (BioRad Laboratories, Hercules, CA, USA) as the standard. Approximately 2 µg of soluble protein was digested using Gold MS-grade trypsin (Promega, San Luis Obispo, CA, USA) at 37 °C for 18 h. The digest was terminated by adding 0.1% (*v*/*v*) formic acid (FA), and the samples were dried under vacuum. Tryptic peptides were dissolved in 0.1% (*v*/*v*) trifluoroacetic acid (TFA) in 2% (*v*/*v*) acetonitrile (ACN) desalted using C_18_ spin cartridges (ThermoFisher Scientific) and eluted using 0.1% (*v*/*v*) FA in ACN, vacuum dried, and stored at −20 °C.

### 2.7. LC–MS Analysis

Mass spectrometry analysis of 2 µg tryptic peptides was performed using a Q Exactive hybrid quadrupole-Orbitrap mass spectrometer (ThermoFisher Scientific) coupled with a nanoflow NLeasy1000 HPLC unit (Thermofisher Scientfic). All 3 biological replicates from the TXS- and CXS-treated bacterial samples were analyzed, creating raw MS files for further analysis. Peptides were separated on an in-house packed C_18_ column (5 μm particles, 300 Å pores, 10 cm) eluting at 300 nL/min. A 2% (*v*/*v*) ACN to 80% ACN (*v*/*v*) gradient in 0.1% (*v*/*v*) FA was used for HPLC, with a total run time of 2 h. After acquisition, spectra were converted to Mascot Generic Format (MGF) files using Mascot Distiller v2.0: (Matrix Science, London, UK) and queried against the *C. nebraskensis* genomic sequence using the Mascot search engine v2.4 (Matrix Science). The following parameters were applied: trypsin with 1 missed cleavage permitted, fixed modification of carbamidomethyl on Cys, variable modifications of oxidation on Met, and deamidation on Asn or Gln. The precursor ion tolerance was ±5 ppm, and MS/MS tolerance was ±0.1 Da. Decoy searches of reversed sequences was enabled. The spectra acquired during all of the LC–MS runs were deposited to the ProteomeXchange Consortium via the PRIDE partner repository [21], with the dataset identifier PXD014510.

### 2.8. LC–MS Data Analysis

Data from LC–MS were loaded into Scaffold (v4.8.6: Proteome Software Inc., Portland, OR, USA). Scaffold normalizes the runs by multiplying each spectrum count in each sample by the average count over total spectrum count of the replicate. Scaffold v4.8.6 (Proteome Software Inc.) was used to validate MS/MS-based peptide and protein identifications. Peptide identifications were accepted if they could be established at greater than 5% probability to achieve a false discovery rate (FDR) less than 0.5% by the Peptide Prophet algorithm [22], with Scaffold delta-mass correction. Protein identifications were accepted if they could be established at greater than 95% probability and contained at least 2 identified peptides. Protein probabilities were assigned by the Protein Prophet algorithm [23]. Proteins that contained similar peptides and could not be differentiated on the basis of MS/MS analysis alone were grouped to satisfy the principles of parsimony. Proteins sharing significant peptide evidence were grouped into clusters. The differential abundant proteins were statistically analyzed using Fisher’s exact test with probability ˂0.05 based on total spectra counts.

### 2.9. Cellulase Activity Assay

Total cellulase activity assay was carried out as previously described [24] on the concentrated Cn filtrates with minor modifications. Where necessary, different Cn filtrate volumes (250, 500, and 1000 µL) were adjusted to 1 mL with filtered xylem sap and added to 0.5 mL of 1% carboxymethyl cellulose (CMC) as a substrate. The mixture was incubated at 40 °C in a water bath for 30 min. Thereafter, reactions were stopped by adding 3 mL of 3,5-dinitrosalicylic acid reagent (DNS) and then the color was developed by placing the samples in boiling water for 5 min. Reaction mixtures were placed in ice-cooled water to quench the reaction, and then 20 mL distilled water was added to each mixture. Absorbance was measured with the Harvard Biochrom Ultrospec 2100 pro UV/Visible Spectrophotometers (ThermoFisher Scientific) at 540 nm against the blank. The experiment was repeated once with 3 replications each. A standard curve was prepared using 0.2% (*w*/*v*) glucose.

### 2.10. Protease Activity Assay

Total protease activity was measured in the concentrated filtrates using casein as substrate [25]. In brief, different Cn filtrate volumes (250, 500, and 1000 µL) were adjusted to 1 mL with the enzyme solution (10 mM sodium acetate buffer with 5mM calcium, pH 7.5) to make a final volume of 1 mL. The assay was carried out by using 5 mL casein solution equilibrated at 37 °C in a water bath, and then 1 mL filtrate concentrates was added to casein solution and incubated at 37 °C for 10 min. After incubation, 5 mL TCA solution was added to stop the reaction. The final volume in the blank tubes was adjusted by adding 1 mL of the enzyme solution. Samples were incubated at 37 °C for 30 min in a water bath, then filtered using a 0.45 µm filter. Color was developed by adding 2 mL sample mixture with 5 mL sodium carbonate solution and 1 mL Folin’s reagent. Samples were incubated at 37 °C for 30 min in a water bath, then filtered using 0.45 µm filters. The absorbance of the developed blue color was measured with the Harvard Biochrom Ultrospec 2100 pro UV/Visible Spectrophotometers (ThermoFisher Scientific) at a wavelength of 660 nm. The activity was calculated on the basis of a standard curve generated using l-tyrosine (Sigma-Aldrich, St. Louis, MO, USA). The experiment was carried out with 3 biological replications and the entire experiment was repeated once.

### 2.11. Statistical Analysis

The analysis was performed on data collected from 3 biological replicates (*n* = 3) for each strain. Standard curves were used to convert spectra values into concentrations for total cellulases and proteases activities, while the disease severity scale values were converted into a percentage prior to statistical analysis. Statistical analysis was performed by the Statistical Analysis Software (SAS) (SAS Institute, Cary, NC, USA; release 9.1) using PROC MIXED module. Data were tested for normality using PROC UNIVARIATE and outliers were removed using Lund’s test [26] (Lund, 1975). The mean values were separated by least squared means and letters assigned by the macro PDMIX800.sas [27] (Saxton, Tadcaster, UK, 1998) with α = 0.05.

## 3. Results

### 3.1. Aggressiveness Assay

The total area under the disease progress curve (total AUDPC) of both disease severity and lesion length (Appendix A) indicated that the Cn strain Cn14-5-1 was significantly more aggressive than strain DOAB232. Strain Cn14-5-1 induced significant damage to corn leaves beginning with developing water-soaked spots, which later turned necrotic. Strain DOAB232 induced significantly low damage, limited to a few millimeters in size, from the inoculation site, leaving the rest of the inoculated leaves undamaged (Appendix A).

### 3.2. Characterisation of Cn Secretome by LC–MS Analysis

The total secretome profile of Cn strains Cn14-5-1 and DOAB232, grown in CXS and TXS, is shown in Appendix A. A total of 745 proteins were identified in the bacterial secretomes for both strains in both xylem sap media (Appendix A). A total of 645 proteins were observed in both sap media with only 57 and 43 proteins identified in CXS and TXS, respectively, for the two Cn strains (Figure 1A). On the other hand, in sap media, a total of 380 proteins were detected between the two Cn strains, with 332 and 33 proteins uniquely recorded by Cn14-5-1 and DOAB232, respectively (Figure 1B). Using the BLAST2GO algorithm, we categorized these secreted proteins into 12 cellular functional processes involved in protein metabolism (30.6%), cell wall metabolism, protein folding, material transporters, nucleic acid metabolism, carbohydrate metabolism, lipid metabolism, stress response, secondary metabolites, various functions, and a group of proteins with unknown function (Figure 1C).

Table 1 shows the number of proteins on the basis of their status in the different secretome pools—presence/absence and uniquely or differentially abundance. The total identified proteins of Cn14-5-1 were approximately double those of DOAB232 in both sap media. Strains Cn14-5-1 and DOAB232 responded differently to CXS and TXS. The secretome patterns of strain Cn14-5-1 in both CXS and TXS were similar while strain DOAB232 exhibited more secreted proteins in CXS. Proteins that were identified only in CXS could more likely contribute to the bacterial colonization and/or pathogenicity on corn rather than those identified in both sap media.

### 3.3. Abundance of Secreted Proteins in Xylem Sap Media

Proteins with various abundance levels were identified in the secretome of the Cn strains in the xylem sap media. Differentially or uniquely abundant proteins were identified in each strain under each sap medium. A total of 269 and 459 proteins were identified only in Cn14-5-1, the highly aggressive strain, under CXS (Appendix A) and TXS (Appendix A), respectively. These include well-characterized protein families involved in carbon metabolism (e.g., glucose-6-phosphate dehydrogenase), protein degradation (e.g., peptidases), and antibiotic resistance (e.g., putative multidrug export ABC transporters as well as a CelA-like protein, a known pathogenicity trait) (Appendix A). Further, a total of 270 and 200 proteins were differentially abundant in CXS (Appendix A) and TXS (Appendix A), respectively. Examples of uniquely and differentially abundant proteins secreted in both media are presented in Table 2 and Table 3, respectively, and were shown to exhibit the presence of a wide range of functional groups such as stress response, pathogenicity, protein degradation/folding, virulence, and carbohydrate metabolism. The majority of the unique proteins in Cn14-5-1 strain were identified under CXS conditions (Table 2). Some proteins with cellulase activity exhibited increased abundance in Cn14-5-1 under CXS, e.g., secreted cellulase, and cellulase-binding and expansin domain-containing proteins (Table 3). Unlike Cn14-5-1, only 40 and 10 secreted proteins were uniquely observed for DOAB232, the low aggressive strain, in CXS (Appendix A) and TXS (Appendix A), respectively. The number of differentially abundant proteins secreted in both sap media by DOAB232 also showed low counts of 40 and 18 for CXS (Appendix A) and TXS (Appendix A), respectively. The majority of uniquely abundant proteins of DOAB232 strain was identified in CXS, which mainly belonged to stress response proteins. A few proteins, such as endo-1,4-beta-xylanase and secreted serine protease, which are suggested to be potential pathogenicity factors, were identified in both sap media (Table 4). Several proteins that were differentially abundant in either sap media belonged to stress response and pathogenicity groups, e.g., 10 Kb chaperonin, serine protease, and endo-1,4-beta-xylanase (Table 5). 

Figure 2 shows the fold change patterns of key stress-related protein families of the two Cn strains treated with CXS and TXS media. Figure 2A shows the changes in 12 secreted proteins that are related to signal perception and transduction. All the secreted proteins exhibited an increment with the exception of the anti-sigma factor, which showed a differentially reduced abundance in strain Cn14-5-1. RNA degradosome polyphosphate kinase and glucokinase (putative catabolite repressor) exhibited the highest fold changes in both xylem sap media for the highly aggressive strain (Figure 2B). The weakly pathogenic strain (DOAB232) either showed no fold change or a reduced abundance for all the 12 secreted proteins. With respect to the 10 protein folding enzymes analyzed, significant increases were observed only in 3 (60 kDa chaperonin, translation initiation factor TF-2, and chaperone protein dnaK) for strain Cn-14-5-1, while DOAB232 had significant increases only for FKBP-peptidyl-polyl-cis-transisomerase in both xylem sap media (Figure 2B).

In addition, proteins with predicted antioxidant activities were identified in both strains when treated with both xylem sap media (Figure 2C). Of the 14 secreted proteins, 9 differentially increased in both xylem sap media for strain Cn14-5-1, while none exhibited significant increase for DOAB232 (Figure 2C). These include catalase (KatA), transketolase tktA, oxidorectuxtase, manganese catalase, organic hydroperoxide resistance protein (Ohr), and thioredoxins differentially increased abundance levels in both xylem sap media in Cn14-5-1 (Figure 2C).

Figure 2D shows the changes in patterns of 12 secreted proteins reported to have potential virulence activities. Strain Cn14-5-1 exhibited significant fold increases in 7 (OpcA protein, NUDIX hydrolase, porphobilinogen deaminase, secreted lipoprotein, Cupin domain-containing protein, chloride anion channel, and phospo-sugar mutase manB) of the 12 virulence-related proteins in CXS and TXS (Figure 2D). On the other hand, fibronectin type III domain-containing protein (putative RTX toxin) was the only virulence-related protein out of 12 that exhibited a differentially increased abundance in DOAB232 in CXS (Figure 2D). PASTA domain-containing protein was the only secreted protein that showed moderate (about 1.5-fold change) in both xylem sap media by strain DOAB232 (Figure 2D).

Different cell wall-degrading enzymes (CWDEs), belonging to four glycosyl hydrolase families (GH3, GH5, GH18, and GH26), varied in abundance in the Cn secetome (Figure 3). However, the abundance levels of the proteins with hydrolytic activity were higher in the secretome of the Cn14-5-1 than DOAB232 (Figure 3A). Significant fold changes were recorded in 10 of the 13 secreted proteins detected in the secretome of strain Cn14-5-1 in both sap media (Figure 3A). The 10 proteins included cellulase, beta-galactosidase, glycosyl hydrolase, glucan debrabching enzyme (GigX), CelA, and beta-glucosidase (Figure 3A). On the other hand, the weakly aggressive strain (DOAB232) exhibited significant positive changes for only endo-1,4-β-xylanases anddetected in both sap media (Figure 3A).

Two classes of peptidases, serine peptidase and metallopeptidase families, with seven different peptidase families from each class, were identified in the Cn secretomes (Figure 3B). In general, the majority of the proteins with proteolytic activity were detected in the secretome of the Cn14-5-1 strain (Figure 3B). Different protein peptidases were observed in the secretome of strain Cn14-5-1 only. However, some aminopeptidase classes belonging to various subfamilies, and metallopeptidase were differentially abundant in Cn14-5-1 (Figure 3B). Some serine protease peptidase family S8A such as secreted subtilase B (sbtB) were differentially abundant only in DOAB232 in both xylem sap media (Figure 3B).

### 3.4. Hydrolytic Enzyme Activity Assays

Cellulase activity in the Cn14-5-1 strain increased significantly with increasing volume of CXS filtrates, which ranged from 2–4 units·mL^−1^, while the cellulose activity in the filtrate of the DOAB232 strain showed significantly lower increases in activity (Figure 4A). A similar trend was observed for protease activity with strain Cn14-5-1, showing significantly higher activity of 0.3 to 0.9 units·mL^−1^ compared to 0.09 to 0.4 units·mL^−1^ for DOAB232 (Figure 4B). Both strains showed increases in protease activity with increasing filtrate volumes (Figure 4B).

### 3.5. Hydrogen Peroxide and Superoxide Assays

Reactive oxygen species, hydrogen peroxide (H_2_O_2_), and superoxide (O_2_^−^) were detected using DAB and NBT, respectively, and localized in corn leaves infected with Cn strains Cn14-5-1 and DOAB232. They were compared to non-infected controls, which exhibited very low staining. Leaves infected with either Cn strains showed deep brown and blue staining for H_2_O_2_ and O_2_^−^, respectively, around the inoculation sites (Appendix A), suggesting the host cell response by the production of reactive oxygen species.

## 4. Discussion

This study profiled the secretomes of two Cn strains that differ in the level of aggressiveness or pathogenicity to corn. This is the first report of the use of LC–MS analysis to investigate the secreted proteins of the corn pathogen, *C. nebraskensis*, and the data presented here will enable a better understanding of the pathogenicity traits in Cn. The generated proteomic data would be a step forward to refine the annotation of the sequenced genome of *C. nebraskensis*. A recent study conducted by Peritore-Galve and colleagues, investigated the implement of proteome peptide sequences to refine the genome annotations of the *Clavibacter michiganensis* susp. *michiganensis*, the causal agent of bacterial canker of tomato, to create a system called proteogenomics [28].

The discrepancies in aggressiveness between Cn14-5-1 and DOAB232, which presented different symptoms clearly on corn, and resulted in different levels of plant damage, provided a platform to study, in vitro, Cn aggressiveness at the protein level using natural xylem sap media. Strain Cn14-5-1 was isolated in 2014, while DOAB 232 was isolated over 40 years ago. Agarkova and colleagues [9] indicated that strain DOAB 232 was pathogenic by direct injection of the bacterial suspension into corn stem. In our study, we used minimal foliar abrasion method, but only Cn14-5-1 inflicted typical symptoms of the Goss’s disease on corn. The differences in aggressiveness could be due to a combination of factors including the effect of long-term storage on DOAB 232 as well as changes at the genome level. Comparative genomics analysis of DOAB 232 and two strains, DOAB395 and DOAB397, isolated at the same year with Cn14-5-1 in 2014, revealed a proteome homology of only 92%, an indication of potential differences in genome arrangements [29]. The LC–MS data presented here contribute to the understanding of this genome-level differences and may explain the variation in aggressiveness of the two Cn strains.

This study identified secreted proteins reported to be involved in potential virulence of Cn strains in response to the corn xylem sap (CXS, host plant) and the tomato xylem sap (TXS, non-host plant) as an alternative medium. Tomato xylem sap (TXS) was used as an alternative induction medium because tomato is a host for *C. michiganensis* subsp. *michiganensis* that does not infect corn plants. The use of synthetic media such as minimal media or rich media, e.g., NBY, would either stress Cn or produce high background noise that might impair reliable mass spectrometric analyses, respectively. We hypothesized that Cn proteins secreted in equal abundance in both CXS and TXS were less likely to be directly involved in the pathogenicity of corn. The identified Cn secreted proteins were assigned into five different categories as follows: adaptation to new ecological niches, signal perception and transduction, stress-responsive proteins, virulence-related proteins, and hydrolytic enzymes for cell wall or protein degradation.

Degradation of plant cell walls is usually a major part of plant pathogen infections [30,31], since it can be a critical step in establishing a successful infection. Plant pathogens, including bacteria, secrete cell wall-degrading enzymes (CWDEs) to break down the cell walls of host plants, which releases nutrients and promotes colonization [30,32,33].

We identified different CWDEs belonging to four glycosyl hydrolase families (GH3, GH5, GH18, and GH26) showing variation in abundance. These enzymes are involved in pectin, hemicellulose, and cellulose hydrolysis [15], and could be required for Cn to colonize corn tissues resulting in the establishment of *C. nebraskensis* secreted cellulase (CelA, GH5), β-glucosidase (GH3), β-galactosidase (GH5), and chitinase (GH18), which hydrolyze cellulose, pectin, and chitin, and were found in Cn14-5-1 grown in CXS. In contrast, only β-1,4-xylanase was identified in DOAB232, which could contribute to its lower aggressiveness and milder symptoms of this strain. In the potato–*Clavibacter sepedonicus* system, cellulases (*cel*A and *cel*B) and xylanases were reported to be upregulated [34]. The putative plasmid-encoded *cel*A gene was also implicated in the pathogenicity of *C. sepedonicus* strains to eggplant [35,36], and concluded that cellulase is a major plasmid-borne virulence determinant [36]. It is unclear whether the lack of plasmid in strain DOAB232 may contribute to its low aggressiveness. The identification of secreted CelA in extracts of Cn14-5-1 in CXS suggests the presence of a plasmid or an unknown production pathway. Some Cn strains contain single large plasmids with a proximate size of 70 kb, which have shown no correlation with Cn strain pathogenicity [37]. Further research is required to determine the involvement of any such plasmid in the virulence of Cn14-5-1. However, our results clearly showed that Cn14-5-1 is a very aggressive strain. The watery-soaked spots seen in symptomatic infected corn leaves infected by Cn14-5-1 suggested the efficiency of the CWDEs and can be considered as direct evidence of the role of CWDEs as virulence factors in this system. Furthermore, the high activity of total cellulase in Cn14-5-1 compared to DOAB232, measured in the filtrates of Cn grown in CXS, provided additional support to the assumption that CWDEs are involved in Cn aggressiveness.

Establishing a successful infection relies not only on degrading host plant cell walls but also on neutralizing host protein-based defense factors. We identified proteins belonging to two peptidase families in Cn14-5-1 and DOAB232. Secreted peptidases are frequently implicated in overcoming host protein defense components, and as such a large arsenal of such enzymes could lead to higher aggressiveness [38]. Secreted proteases are used by bacterial pathogens to evade plant immune systems and maintain homeostasis [39]. CXS medium boosted the secretion of proteases, e.g., subtilase C (sbtC, S8A) and subtilase B (subtB, S8A), especially in Cn14-5-1, which secreted more proteases and peptidases than DOAB232. Similar proteolytic enzymes, specifically from the serine proteases families Ppa, Sbt. and Chp, were identified in tomato infected by *C. michiganensis* subsp. *michiganensis* [15]. The authors concluded that these enzymes were likely targeting host immune components as well as structural components to facilitate disease progression. The protease assay indicated significantly higher levels of proteases in Cn14-5-1, which may contribute to degradation of plant defense proteins and promote colonization. A similar conclusion was reported on the involvement of proteases in colonization of *Clavibacter michiganensis* and *Clavibacter sepedonicus* [3]. An important next step will be to find targets for these enzymes, as these would be host factors potentially involved in resistance to Goss’s wilt disease. Besides secreted hydrolytic proteins, other secreted proteins with various cellular functions might also be important in Cn–host interaction.

The ability of bacteria to utilize carbohydrates from the surrounding environment is a positive sign of adaptation. Unlike DOAB232, secretion of carbohydrate metabolism-related proteins was obvious in strain Cn14-5-1 treated with CXS. Furthermore, Cn14-5-1 secreted some carbohydrate metabolism-related proteins in TXS medium, an indication of the ability of Cn14-5-1 to utilize carbohydrates from non-host plants. Some of these identified proteins, e.g., glucokinase (GLK) and glycerol kinase, function as transcriptional regulators or catalytic enzymes for carbohydrate metabolism, respectively [40,41,42]. Identifying these proteins in the highly aggressive Cn14-5-1 strain but not in DOAB232 provided further evidence that this isolate utilizes host carbon resources.

Activity of transcriptional and translational machineries in bacterial cells are essential for cellular perception and signal transduction. The ability to perceive and transmit signals is crucial to prepare and respond to plant defense factors. The majority of transcriptional regulators and DNA-binding proteins were identified in Cn14-5-1 treated with CXS, e.g., antibiotic resistance regulator (MarR) protein family, a one-component signal transduction [43], transcriptional regulator (TetR) [44,45], and ROK family transcriptional regulator [46,47], may provide evidence of the efficiency of strain Cn14-5-1 to recognize and respond to plant stimuli.

Furthermore, phosphorylation is a crucial process that modulates protein activities. Secreted proteins with high abundance in Cn14-5-1 that are involved in protein phosphorylation, e.g., serine/threonine protein phosphatase, HAD family phosphatase, and NTP phosphatase, when treated with CXS may indicate the preference to the host stimuli.

The Cn–corn interaction occurs in the apoplast and extracellular spaces, where bacterial cells secrete virulence factors to evade plant defense factors and promote infection. These secreted virulence factors are translocated via transporter proteins [15,48]. We identified a larger number of transporters in Cn14-5-1 than DOAB232, which could be another potential piece of evidence to explain variations in Cn strain aggressiveness. Plant pathogens negatively affect the integrity of host plant cell walls and cell membranes to cause cell death and release nutrients [49]. Thereafter, pathogens employ different strategies to take up these nutrients, including passive diffusion and active uptake. Various types of membrane transporters facilitate the latter, e.g., ABC transporters [50,51] and ATP-binding cassette ABC transporters [52].

Under stress conditions, e.g., environmental changes, bacterial cellular proteins are subjected to alterations. Chaperones are responsible for folding either newly synthesized polypeptides or refolding stress-denatured proteins, or both [53,54]. The secretome of Cn14-5-1 had various types of chaperones with different levels of abundance, e.g., peptidyl-poly cis/trans isomerase (PPIase), which reacts with the chaperone DnaK system to fold newly synthesized polypeptides [55]. In addition, 60 kDa chaperonin and 10 kDa chaperonin (GroES) are essential in newly synthesized polypeptides, as well as stress-denatured proteins [56,57]. Abundance of these types of chaperones in Cn14-5-1 treated with CXS may demonstrate efficient re-use of resources, which may enhance its ecological fitness leading to high aggressiveness.

Phytopathogens cope with host plant reactive oxygen species (ROS)-based defenses [58] by secreting antioxidant enzymes. Corn plants accumulate ROS in the infection site to restrict bacterial growth and to hinder disease progress. High abundance of secreted ROS scavenging proteins in Cn14-5-1 secretome, e.g., catalase (KatA), and manganese catalase were deployed to mitigate the deleterious effect of ROS as the first host plant chemical defense barrier [59]. Finally, lipoproteins [60,61], cupin domain-containing proteins [62], and Stk1 family PASTA domain-containing Ser/Thr kinase [63] were also observed in Cn14-5-1, and these may also contribute to the higher aggressiveness of this strain.

## 5. Conclusions and Future Perspectives

Host and non-host xylem sap represent a convenient and biologically relevant model for culturing Cn and performing comparative proteomics of the Cn–corn interaction in vitro. We found that the pathogen secretome was altered in response to host xylem sap compared to non-host xylem sap. We identified significant changes in the abundance of key proteins implicated in pathogenicity. Since these changes are in response to host xylem, it is likely that these proteins could play important roles in the corn-pathogen interaction through responding to chemical stimuli present in the host xylem sap. The xylem sap system has the advantage that the host proteome is very minute from the protein extracts. This minimizes background noise from LC–MS data and greatly simplifies downstream proteomic analyses. A comparison of the secretomes of weakly and highly aggressive strains of Cn indicated that cell wall-degrading enzymes, proteases, and other types of proteins might have important roles in Goss’s wilt disease establishment. Further experiments on Cn gene knockout for CWDEs and/or proteases encoding genes will be useful in deciphering the potential role of these two major secreted proteins in Cn aggressiveness.

## Figures and Tables

**Figure 1 proteomes-09-00001-f001:**
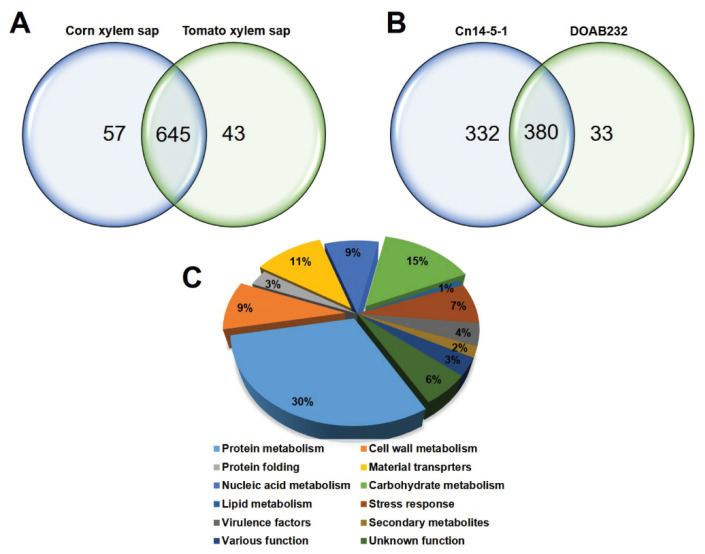
Categories of the identified secretome proteins of Cn strains and their abundance. (**A**) Venn diagram shows unique and differentially abundant proteins under corn xylem sap (CXS) and tomato xylem sap (TXS) media. (**B**) Venn diagram shows unique and differentially abundant proteins of Cn14-5-1 and DOAB232 strains. (**C**) Different classes and the abundance of the identified protein shown on the basis of the predicted biological function.

**Figure 2 proteomes-09-00001-f002:**
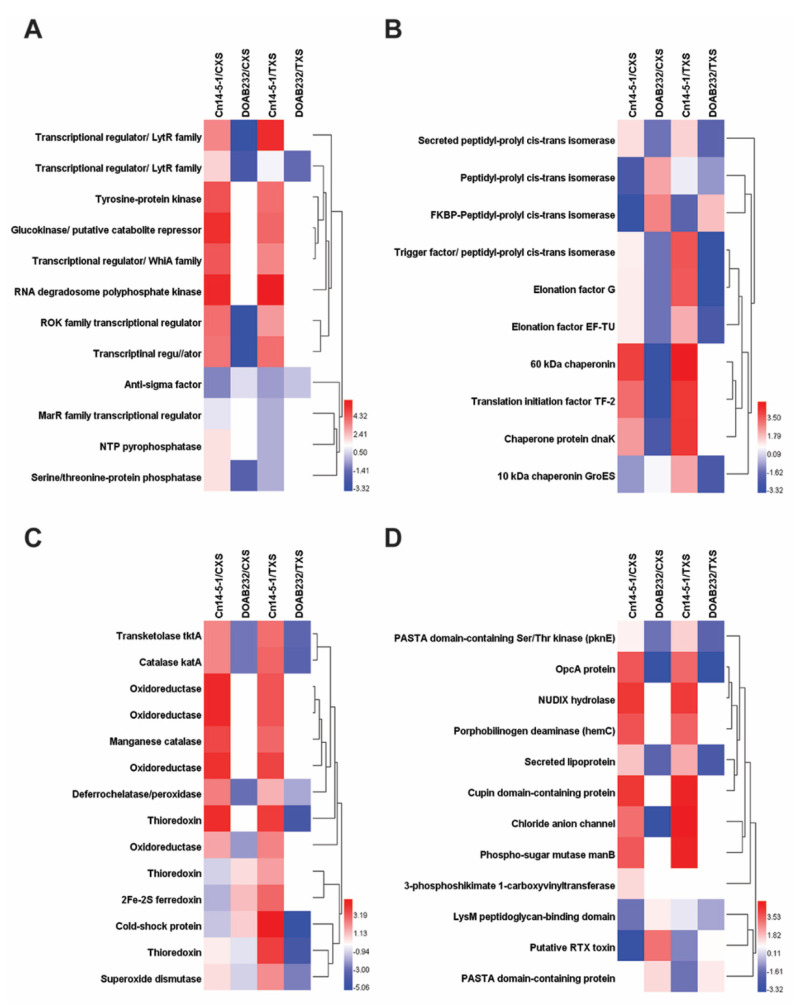
Stress-related proteins patterns of the two strains of *Clavibacter nebraskensis* treated with different xylem sap media. Secretome profile of Cn14-5-1 was normalized by that of the DOAB232 strain, and vice versa in each sap medium to identify the differential stress-related protein profile under corn xylem sap (CXS) and tomato xylem sap (TXS) induction. (**A**) Signal perception and transduction; (**B**) protein folding; (**C**) reactive oxygen species (ROS) scavenging proteins; and (**D**) virulence-related proteins. Hierarchical clustering was performed using the group average (unweighted pair-group method (UWPGM)) method.

**Figure 3 proteomes-09-00001-f003:**
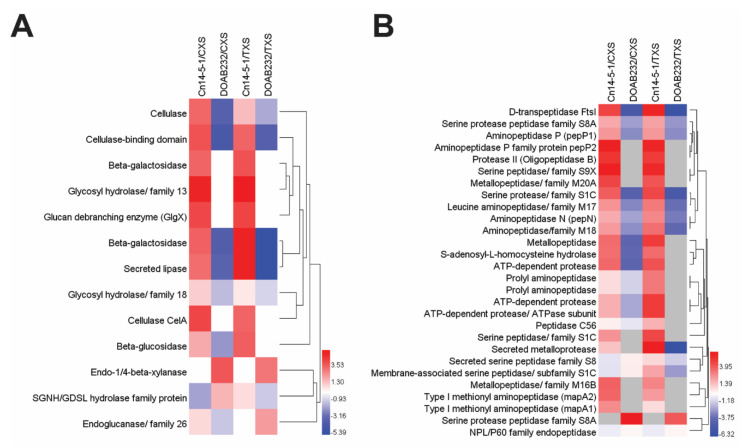
Hydrolytic proteins patterns of the two strains of *Clavibacter nebraskensis* treated with different xylem sap media. Secretome profile of Cn14-5-1 was normalized by that of the DOAB232 strain, and vice versa in each sap medium to identify the differential stress-related protein profile under corn xylem sap (CXS) and tomato xylem sap (TXS) induction. (**A**) Cell wall-degrading enzymes, (**B**) proteases. Hierarchical clustering was performed using the group average (unweighted pair-group method (UWPGM)) method.

**Figure 4 proteomes-09-00001-f004:**
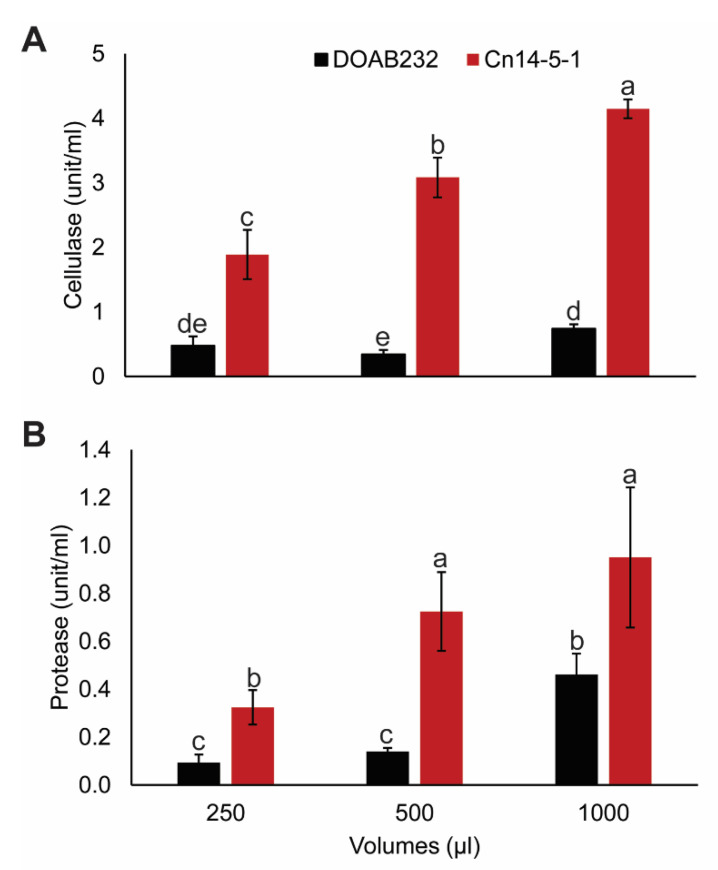
Enzymatic activity of secreted cellulases and proteases in filtrates of *Clavibacter nebraskensis* strains. The weakly aggressive DOAB232 strain in black and the highly aggressive Cn14-5-1 strain in red. (**A**) Enzymatic activity of total cellulase, and (**B**) protease were assayed in Cn filtrates. The mean values (*n* = 3) were separated by least squared means, and values with the same letter are not significantly different at (*p* < 0.05).

**Table 1 proteomes-09-00001-t001:** Total secreted proteins in Cn isolates under different sap media.

Identified Proteins	Corn Xylem Sap (CXS)		Tomato Xylem Sap (TXS)	
	Cn14-5-1	DOAB232	Cn14-5-1	DOAB232
Total proteins	658	396	678	229
Unique proteins	269	40	459	10
Differentially abundant	270	40	200	18

(CSX) corn xylem sap, (TXS) tomato xylem sap.

**Table 2 proteomes-09-00001-t002:** Proteins identified only in Cn14-5-1 in the different sap media.

Accession No.	Identified Proteins	MW (KDa)	Gene/Locus	Type of Sap		Functional Group
				CXS	TXS	
CCE76040.1	Serine peptidase, family S9X	76	CMN_02099	+	−	Protein degradation
CCE75111.1	pepP2 protein	57	pepP2	+	−	Protein degradation
CCE76391.1	Polyphosphate kinase	82	ppK	+	−	Protein phosphorylation
CCE75799.1	Glucokinase	33	glkA	+	−	Catabolite repressor
CCE74648.1	Radical SAM domain-containing protein	46	CMN_00689	+	−	Stress response
CCE76022.1	Protease II (oligopeptidase B)	80	ptrB	+	−	Protein degradation
CCE75620.1	Metallopeptidase, peptidase family M20A	47	CMN_01675	+	−	Protein degradation
CCE74827.1	Putative tyrosine-protein kinase	49	CMN_00871	+	−	Stress response
CCE74652.1	Oxidoreductase	35	CMN_00693	+	+	Stress response
CCE75671.1	Transcriptional regulator, WhiA family	35	whiA	+	−	Signal transduction
CCE74496.1	Thioredoxin	21	ccbD	+	−	Stress response
CCE75993.1	Oxidoreductase	36	CMN_02051	+	+	Stress response
CCE75476.1	NUDIX hydrolase	19	CMN_01528	+	+	Virulence
CCE76025.1	Cupin_2 domain-containing protein	32	CMN_02083	+	+	Virulence
AHN16207.1	Cellulase	78	CelA	+	+	Pathogenicity
CCE75945.1	Metallopeptidase, family M16B	48	pepR	+	−	Protein degradation
CCE76486.1	Methionine aminopeptidase	29	mapA2	+	−	Protein degradation
CAN00621.1	Porphobilinogen deaminase	34	hemC	+	+	Pathogenicity
AJW80248.1	Manganese catalase	31	DZF93_04220	+	+	Stress response
CCE74906.1	ManB protein	61	manB	+	+	Pathogenicity
CCE74104.1	Putative glycosyl hydrolase, family 2	69	CMN_00123	+	−	Pathogenicity
AJW78817.1	ATP-dependent Clp protease proteolytic subunit	21	clpP	+	+	Stress response
CCE76317.1	Serine peptidase, family S1C	51	CMN_02381	+	−	Protein degradation
CCE75886.1	NTP pyrophosphatase	22	CMN_01943	+	−	Signal transduction
CCE74756.1	Putative secreted metalloprotease	46	CMN_00799	+	−	Protein degradation
CCE76463.1	Alanine racemase/kinase fusion protein	59	alr2	+	−	Signal transduction
AJW79024.1	HAD family phosphatase	25	VO01_07690	+	−	Signal transduction
AJW80208.1	MarR family transcriptional regulator	17	DZF93_09435	+	−	Signal transduction
CCE74692.1	Chloride anion channel	25	CMN_00734	−	+	Pathogenicity
CCE74774.1	ATP-dependent protease, ATPase subunit	92	clpC	−	+	Protein degradation
CCE74648.1	Radical SAM domain-containing protein	46	CMN_00689	−	+	Stress response
CCE76686.1	Alpha-glucosidase, glycosyl hydrolase family 13	64	aglC	−	+	Pathogenicity
CCE75391.1	ATP-dependent protease, proteolytic subunit	25	clpP2	−	+	Stress response
CCE76068.1	1-Deoxy-D-xylulose 5-phosphate reductoisomerase	38	dxr	−	+	Pathogenicity
CCE75358.1	Glucan debranching enzyme	82	treX	−	+	Pathogenicity
CCE74827.1	Putative tyrosine-protein kinase	49	CMN_00871	−	+	Stress response
CCE76839.1	2-Keto acid dehydrogenase	41	CMN_02915	−	+	Energy production
CCE74969.1	Lipoprotein	59	lpqB	−	+	Pathogenicity
CCE74801.1	Beta-glycosidase	54	bglJ	−	+	Pathogenicity
CCE76286.1	Alpha glycosidase	81	CMN_02350	−	+	Pathogenicity
AJW79410.1	Sugar kinase	32	VO01_09940	−	+	Signal transduction
CCE74250.1	Alpha-L-arabinofuranosidase	55	abfA1	−	+	Pathogenicity
AJW78269.1	Organic hydroperoxide resistance protein	14	Ohr	−	+	Stress response

(CSX) corn xylem sap, (TXS) tomato xylem sap, (+) presence, (−) absence.

**Table 3 proteomes-09-00001-t003:** Differentially abundant proteins in Cn14-5-1 in different sap media.

Accession No.	Identified Proteins	MW (KDa)	Gene/Locus	Fisher’s Exact Test	Type of Sap		Fold Change	Functional Group
				*p*-Value (*p <* 0.05)	CXS	TXS		
CCE76069.1	Secreted peptidyl-prolyl cis-trans isomerase	34	CMN_02128	0.00013	−	+	3.2	Protein folding
ALD12817.1	Cold-shock protein	7	AES38_07750	0.00018	−	+	38	Stress response
CCE74756.1	Putative secreted metalloprotease	46	CMN_00799	0.00029	−	+	80	Protein degradation
CCE75860.1	Secreted lipase	30	CMN_01917	0.00047	+	−	10	Pathogenicity
CCE76016.1	Beta-galactosidase, lactase	113	lacZ	0.001	−	+	42	Pathogenicity
CCE75767.1	Metallopeptidase	49	CMN_01822	0.0016	+	−	14	Protein degradation
CCE74047.1	Chaperone protein dnaK	67	dnaK	0.0017	+	−	6.1	Protein folding
CCE74654.1	Catalase	57	katA	0.0018	+	−	5.8	Stress response
CCE75796.1	Non-specific serine/threonine protein kinase	69	pknE	0.0051	−	+	3.3	Virulence
CCE75569.1	Leucine aminopeptidase, family M17	52	pepA	0.006	+	−	6.9	Protein degradation
CCE74438.1	Transcriptional regulator, LytR family	43	CMN_00472	0.009	+	−	13	Virulence
CCE74438.1	Transcriptional regulator, LytR family	43	CMN_00472	0.013	−	+	48	Virulence
CCE75659.1	Transketolase	75	tktA	0.015	+	−	5.8	Stress response
CCE74692.1	Chloride anion channel	25	CMN_00734	0.038	+	−	11	Virulence
CCE75389.1	Peptidyl-prolyl cis-trans isomerase	52	tig	0.043	−	+	14	Pathogenicity
CCE74969.1	Lipoprotein	59	lpqB	0.044	+	−	22	Virulence
CCE74654.1	Catalase	57	katA	0.071	−	+	9.2	Stress response
CCE75600.1	Endo-1,4-beta-xylanase	71	xysB	0.084	−	+	2.5	Pathogenicity
AJW79410.1	Sugar kinase	32	VO01_09940	0.089	+	−	18	Signal transduction
CCE76365.1	60 KDa chaperonin	57	groEL	<0.00010	+	−	19	Protein folding
CCE74122.1	Cellulose-bindingand an expansin domain	37	CMN_00144	<0.00010	+	−	16	Pathogenicity
CCE76326.1	Secreted cellulase	58	cel	<0.00010	+	−	11	Pathogenicity
CCE75788.1	FtsI protein	62	ftsI	<0.00010	+	−	30	Pathogenicity
CCE75200.1	Serine protease, family S1C	50	CMN_01248	<0.00010	+	−	18	Protein degradation
CCE76016.1	Beta-galactosidase, lactase	113	lacZ	<0.00010	+	−	13	Pathogenicity
CCE75860.1	Secreted lipase	30	CMN_01917	<0.00010	−	+	41	Pathogenicity
CCE76365.1	60 KDa chaperonin	57	groEL	<0.00010	−	+	37	Protein folding
CCE75200.1	Serine protease, family S1C	50	CMN_01248	<0.00010	−	+	29	Protein degradation
CCE74047.1	Chaperone protein dnaK	67	dnaK	<0.00010	−	+	22	Protein folding
CCE74122.1	Cellulose-binding and an expansin domain	37	CMN_00144	<0.00010	−	+	12	Pathogenicity
CCE76326.1	Endoglucanase	58	cel	<0.00010	−	+	3	Pathogenicity
CCE76173.1	Secreted serine peptidase family S8	43	CMN_02235	<0.00010	−	+	2.1	Protein degradation
CAQ02078.1	Glucose-6-phosphate 1-dehydrogenase	58	Zwf	0.025	+	−	4.9	Carb. Metabolism
CCE75669.1	Glyceraldehyde 3-phosphate dehydrogenase	36	gapA	<0.00010	+	−	7.9	Carb. Metabolism
CCE74538.1	Putative levansucrase	65	sacB	0.00051	−	+	4.3	Carb. Metabolism
AJW79067.1	OpcA protein	35	DZF93_01230	0.013	+	−	14	Virulence
CCE75796.1	PASTA domain containing Ser/Thr kinase	69	pknE	0.025	+	−	2.3	Virulence
CCE74059.1	Glycosyl hydrolase, (chitinase) family 18	39	CMN_00077	0.0026	+	−	1.6	Pathogenicity

(CSX) corn xylem sap, (TXS) tomato xylem sap, (+) presence, (−) absence.

**Table 4 proteomes-09-00001-t004:** Proteins identified only in DOAB232 in different sap media.

Accession No.	Identified Proteins	MW (KDa)	Gene/Locus	Type of Sap		Functional Group
				CXS	TXS	
CCE75047.1	Putative secreted 5’-nucleotidase	73	CMN_01095	+	+	Nucleotide catabolism
CAQ00484.1	Putative solute-binding lipoprotein	46	CMS0363	+	+	Virulence
AJW80270.1	Integration host factor	10	VO01_15120	+	−	Stress response
CCE74134.1	Rhodanese domain-containing protein	10	CMN_00157	+	−	Stress response
CCE74199.1	Sugar ABC transporter	43	CMN_00224	+	−	Cell surface
CCE75732.1	Rhodanese domain-containing protein	12	CMN_01787	+	−	Stress response
AJW79484.1	Exodeoxyribonuclease VII small subunit	9	xseB	+	−	DNA catabolic
AJW78093.1	General stress protein CsbD	6	DZF93_05670	+	−	Stress response
CCE74980.1	Alkyl hydroperoxide reductase	17	bcp	+	−	Stress response
AJW79401.1	Antibiotic biosynthesis monooxygenase	11	DZF93_04600	+	−	Stress response
CCE74048.1	Heat shock chaperone GrpE	24	grpE	+	−	Protein folding
CCE76425.1	Secreted serine protease, peptidase family S8A	121	sbtB	+	+	Protein degradation
CCE75599.1	Endo-1,4-beta-xylanase	45	xysA	+	+	Pathogenicity
CCE76583.1	Endoglucanase, glycosyl hydrolase family 26	47	CMN_02651	−	+	Pathogenicity
CCE74634.1	Esterase	25	CMN_00675	−	+	Virulence

(CSX) corn xylem sap, (TXS) tomato xylem sap, (+) presence, (−) absence.

**Table 5 proteomes-09-00001-t005:** Differentially abundant proteins in strain DOAB232 grown in the different sap media.

Accession No.	Identified Proteins	MW (KDa)	Gene/Locus	Fisher’s Exact Test	Type of Sap		Fold Change	Functional Group
				*p*-Value (*p <* 0.05)	CXS	TXS		
CCE74709.1	Putative extracellular nuclease	74	CMN_00751	<0.00010	+	−	6.0	Nucleotide degradation
A5CU64.1	10 kDa chaperonin	11	groES	<0.00010	+	−	1.8	Protein folding
ALD12817.1	Cold-shock protein	7	AES38_07750	<0.00010	+	−	2.1	Stress response
CCE76393.1	Phosphate-binding protein PstS	37	pstS	<0.00010	+	−	1.2	Transport protein
CCE76173.1	Secreted serine peptidase family S8	43	CMN_02235	<0.00010	+	−	1.4	Protein degradation
CCE74340.1	Levan fructotransferase	57	CMN_00371	<0.00010	+	−	4.1	Carb. metabolism
CCE76401.1	Putative hydrolase	22	CMN_02466	<0.00010	+	−	3.4	Protein degradation
CCE76077.1	Putative RTX toxin	204	CMN_02136	<0.00010	+	−	10.0	Virulence
CCE76590.1	Sugar ABC transporter	48	CMN_02658	<0.00010	+	−	7.5	Transport protein
CCE75600.1	Endo-1,4-beta-xylanase	71	xysB	<0.00010	+	−	6.0	Pathogenicity
CCE75697.1	FKBP-type peptidyl-prolyl cis-trans isomerase	13	CMN_01752	<0.00010	+	−	8.0	Protein folding
CCE76397.1	Anti-sigma factor	29	CMN_02462	0.0017	+	−	1.8	Stress response
CCE75288.1	Membrane-associated serine peptidase	40	CMN_01337	0.0057	+	−	1.5	Protein degradation
CCE74709.1	Putative extracellular nuclease	74	CMN_00751	<0.00010	−	+	8.4	Pathogenicity
CCE75697.1	FKBP-type peptidyl-prolyl cis-trans isomerase	13	CMN_01752	0.001	−	+	4.0	Protein folding
CCE76165.1	Ferritin-like domain-containing protein	32	CMN_02227	<0.00010	−	+	3.5	Element acquisition
CCE74340.1	Levan fructotransferase	57	CMN_00371	<0.00010	−	+	2.7	Carb. metabolism
CCE75152.1	Agglutinin receptor precursor	48	CMN_01200	<0.00010	−	+	2.5	Signal transduction
CCE76077.1	Putative RTX toxin	204	CMN_02136	<0.00010	−	+	2.0	Pathogenicity
CCE76397.1	Anti-sigma factor	29	CMN_02462	<0.00010	−	+	1.3	Stress response
CCE74757.1	NPL/P60 family endopeptidase	45	CMN_00800	<0.00010	−	+	1.3	Protein degradation
AJW79074.1	Superoxide dismutase	23	SOD	0.027	−	+	5.2	Stress response
CCE76077.1	Putative RTX toxin	204	CMN_02136	<0.00010	+	−	10	Virulence
CCE75795.1	Secreted LysM peptidoglycan-binding protein	43	CMN_01850	<0.00010	+	−	2.4	Virulence

(CSX) corn xylem sap, (TXS) tomato xylem sap, (+) presence, (−) absence.

## Data Availability

Data is contained within the article and the Appendix A.

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
