# Peer review of "Secretome Analysis of Clavibacter nebraskensis Strains Treated with Natural Xylem Sap In Vitro Predicts Involvement of Glycosyl Hydrolases and Proteases in Bacterial Aggressiveness"

_proteomes, 2021, doi:10.3390/proteomes9010001_

Round 1
Reviewer 1 Report
Major Comments to Author:
The Manuscript ID Proteomes-1038505 entitled "Secretome analysis of Clavibacter nebraskensis strains challenged with natural xylem sap predicts involvement of glycosyl hydrolases and proteases in bacterial aggressiveness (Manuscript ID: Proteomes-1038505)" is easy to follow. The authors investigated differences of secreted proteomes between highly (Cn14-15-1) and weakly (DOAB232) aggressive strains of Clavibacter nebraskensis (Cn) of Cn strains to determine the potential factors involved in Cn-host interactions. The results provided the evidences to indicate the putative important proteins involved in Goss’s wilt disease establishment were the cell wall-degrading enzymes and proteases. The analysis of the comparative secretomes seem correct, with appropriate explanation and discussion. The manuscript is well-written in English, the results in this manuscript are worth to be published in Proteomes (ISSN 2227-7382). I think this manuscript is acceptable for publication after minor revision (see suggestions below).
Specific comments:
- I would suggest using L or l for liter in the whole manuscript.
- May change “challenged” to “treated” in the whole manuscript.
- Please improve the picture quality of figures 1 & 2 (indistinct word).
- Please revised the whole References list;
- Ref 55 & 56 were not cited in the manuscript?
Author Response
We would like to thank the reviewer for their comments.
We made all of the changes suggested by the reviewer. Here are our responses point by point:
Reviewer 1.
Major Comments to Author:
The Manuscript ID Proteomes-1038505 entitled "Secretome analysis of Clavibacter nebraskensis strains challenged with natural xylem sap predicts involvement of glycosyl hydrolases and proteases in bacterial aggressiveness (Manuscript ID: Proteomes-1038505)" is easy to follow. The authors investigated differences of secreted proteomes between highly (Cn14-15-1) and weakly (DOAB232) aggressive strains of Clavibacter nebraskensis (Cn) of Cn strains to determine the potential factors involved in Cn-host interactions. The results provided the evidences to indicate the putative important proteins involved in Goss’s wilt disease establishment were the cell wall-degrading enzymes and proteases. The analysis of the comparative secretomes seem correct, with appropriate explanation and discussion. The manuscript is well-written in English, the results in this manuscript are worth to be published in Proteomes (ISSN 2227-7382). I think this manuscript is acceptable for publication after minor revision (see suggestions below).
Specific comments:
I would suggest using L or l for liter in the whole manuscript.
>> Modified as suggested.
May change “challenged” to “treated” in the whole manuscript.
>> Modified as suggested.
Please improve the picture quality of figures 1 & 2 (indistinct word).
>> Modified as suggested.
Please revised the whole References list;
Ref 55 & 56 were not cited in the manuscript?
>>> Removed and number adjusted.
Reviewer 2 Report
Summary
This paper focuses on characterizing the proteome of two Clavibacter nebraskensis (Cn) strains with differing degrees of virulence using xylem sap of different known plant hosts of pathogens within the Clavibacter genus, in particular its known host corn and the host of C. michiganensis, tomato. It provides insights into the small, but notable, differences Cn proteomes can have and how they may explain the variation of virulence on their known hosts. Finally, it sheds light on how alternate plant hosts mat impact gene expression and thus the proteome secreted, leading to different patterns of host colonization and disease. Considering the lack of investigation into these biological processes in a key class of economically import Gram-positive actinobacterial pathogens, this work is a timely addition. However, there are issues with the experimental design and lack of detail provided. Several claims are overstated and not founded based on the data presented. Specific comments are noted below.
Major Comments:
- The title should better represent the work at hand. As-is, the title and abstract indicate that the work was carried out in planta. It should be rephrased to signify that work is performed in vitro.
- One major concern regarding the quantification of peptides found is how the data was normalized before differential abundance was determined. No normalization corrections were described in the methodology and could lead to biases in downstream analyses and conclusions.
- A second major concern is if there were any biological replicates of the LC-MS performed. I assume only one replication since a Fisher’s exact test was used. If there is an n = 1, conclusions should be rephrased/toned down appropriately, the n should be explicitly stated. This is a major limitation of the work, but I don’t see how it can really be corrected at this point. It is difficult to make biological inferences with an n of 1.
- There are multiple formatting errors in Tables 2 through 5.
- The proteomes should be determined using nebraskensis not C. michiganensis.
- What is the growth of both strains after 12h in xylem media from different organisms?
Minor Comments:
- The terms aggressive and weak with respect to virulence likely only relate to the corn genotype tested, which should be tolerant to Cn. This should be specified.
- While it may not impact the findings and overall conclusions, it is unclear why the authors choose to culture the bacterial strains in a lower temperature. Cn is commonly grown at 25°C and many day/night temperatures are around 26-28/22°C (see Mbofung et al., 2016. Phytopathology; Mallowa et al., 2016. Plant Disease; among others).
- This paper (Peritore-Galve et al., 2019. Proteomics) should be listed and discussed due to its relevancy to this work.
- For Figure 1B, which host sap was used? It is not listed in the figure legend nor text and doesn’t match with any of the values in Table 1. Please clarify/fix this.
- Figure 4 – missing information on statistical analyses, error bars, and n. This is a key result and should be explained more clearly in the results or legend.
- Section 3.3 in the results was particularly hard to follow at times. We implore the authors to write this section in a more digestible manner and/or create a figure on the findings.
- Page 6, line 249: fold changes were statistically significant at p<0.0. We presume the authors intended a p-value of 0.05. However if this is not the case, the authors should clarify the cutoff of the value (i.e. 0.001 vs. 0.0001) and why a stringent approach is necessary.
- Figure S1 is missing statistical analyses
- Figure S2 should include quantification of ROS.
- In Table 2, it is unclear why some proteins identified are listed twice with each entry being present in one sap and then the other, whereas other proteins are listed once with presence in both saps (ex. CMN_00689 vs CMN_02051)
- In Table 3, there are notable discrepancies between multiple gene loci identifiers and their protein annotations (ex. CMN_01917: Secreted lipase vs. Serine protease, family S1C). This may be due to formatting issues of the table but should be nevertheless checked and fixed as appropriate.
- For Figure 2, the clustering method should be listed (k-means vs. hierarchical cluster such as Ward’s method, etc.).
- Page 14, line 321: subtilase B (subtB) should be correct as sbtB; similarly on page 16, line 401.
- Page 15, line 353: It is unclear how long-term storage would have any impact on pathogen virulence.
- While the authors focused on proteins which were found wither only in Cn14-15-1 and/or in corn based xylem media, it is important to note that since many, if not most, of the plant targets are unknown.
- Page 16, line 393: while the celluase activity referred to does show increased activity, it does not conclude that they are acting as virulence factors. Rather it shows that bacterial loci that act as virulence genes in other Clavibacter species are functional and have increased activity.
- Page 16, line 406-407: It’s unclear how increased protease activity would lead the authors to conclude ‘…an accelerated action to deplete the native CXS proteins” when no targets of these proteins are known and casein is used as a substrate. This should be toned down.
Author Response
First, we would like to thank the reviewer very much for their valuable comments, to which we responded as listed below (preceded by ">>>"):
Reviewer 2
Comments and Suggestions for Authors
Summary
This paper focuses on characterizing the proteome of two Clavibacter nebraskensis (Cn) strains with differing degrees of virulence using xylem sap of different known plant hosts of pathogens within the Clavibacter genus, in particular its known host corn and the host of C. michiganensis, tomato. It provides insights into the small, but notable, differences Cn proteomes can have and how they may explain the variation of virulence on their known hosts. Finally, it sheds light on how alternate plant hosts mat impact gene expression and thus the proteome secreted, leading to different patterns of host colonization and disease. Considering the lack of investigation into these biological processes in a key class of economically import Gram-positive actinobacterial pathogens, this work is a timely addition. However, there are issues with the experimental design and lack of detail provided. Several claims are overstated and not founded based on the data presented. Specific comments are noted below.
Major Comments:
The title should better represent the work at hand. As-is, the title and abstract indicate that the work was carried out in planta. It should be rephrased to signify that work is performed in vitro.
>>> Secretome analysis of Clavibacter nebraskensis strains treated with natural xylem sap in vitro predicts involvement of glycosyl hydrolases and proteases in bacterial aggressiveness.
- One major concern regarding the quantification of peptides found is how the data was normalized before differential abundance was determined. No normalization corrections were described in the methodology and could lead to biases in downstream analyses and conclusions.
>>> Data from all LC-MS runs were loaded into Scaffold (v4.8.6: Proteome Software, Inc., Portland OR, USA). Scaffold normalizes the runs by multiplying each spectrum count in each sample by the average divided by the replicate’s total spectrum count.
- A second major concern is if there were any biological replicates of the LC-MS performed. I assume only one replication since a Fisher’s exact test was used. If there is an n = 1, conclusions should be rephrased/toned down appropriately, the n should be explicitly stated. This is a major limitation of the work, but I don’t see how it can really be corrected at this point. It is difficult to make biological inferences with an n of 1.
>>> The mass spectrometric data were generated from three biological replications for each isolate in each xylem sap (please, see line 120). Each biological sample was injected twice in the LC-MS. The Fisher’s exact test is the default built-in statistical inference module of the Scaffold Software for use on the sets similar to what was generated in this study. The authors believe the test is valid based on the recommendations of the Scaffold software developers.
- There are multiple formatting errors in Tables 2 through 5.
>>> Adjusted
- The proteomes should be determined using nebraskensis not C. michiganensis.
>>> The C. nebraskensis database was used not C. michiganensis. The spelling error came from changing the name of Clavibacter michiganensis subsp. nebraskensis to C. nebraskensis as a separate Clavibacter species, (Corrected).
- What is the growth of both strains after 12h in xylem media from different organisms?
>>> We adjusted the cell intensity (cfu) to be the same for both strains grew on NBY media before adding xylem sap. We did not measure the growth of Cn strains after incubation in xylem sap media, but we streaked a loop from each strain incubated in each xylem sap on NBY plates overnight to ensure viability of both strains.
Minor Comments:
- The terms aggressive and weak with respect to virulence likely only relate to the corn genotype tested, which should be tolerant to Cn. This should be specified.
>>> The term aggressiveness either highly or weakly refers to the damage induced by Cn strains. Cn14-5-1 induced significant damage, while DOAB232 induced significant low damages, not only in the current corn hybrids by also with other tested commercial and inbred corn hybrids.
- While it may not impact the findings and overall conclusions, it is unclear why the authors choose to culture the bacterial strains in a lower temperature. Cn is commonly grown at 25°C and many day/night temperatures are around 26-28/22°C (see Mbofung et al., 2016. Phytopathology; Mallowa et al., 2016. Plant Disease; among others).
>>> Clavibacter nebraskensis strains can grow in a broad range of temperatures, not limited to a certain temperature. We have used 23-24 ℃ for growth that represent the average temperature in southern Manitoba in the summer time, especially during the rainy season when most of our Cn collection was isolated.
- This paper (Peritore-Galve et al., 2019. Proteomics) should be listed and discussed due to its relevancy to this work.
>>> Added to discussion and references list.
- For Figure 1B, which host sap was used? It is not listed in the figure legend nor text and doesn’t match with any of the values in Table 1. Please clarify/fix this.
>>> 1. Figure 1A represents classification of the total identified proteins in each xylem sap for both strains, while 1B represents classification of the total identified proteins in each Cn isolates in both xylem sap media.
>>> 2. The number in table 1 represents classification of the identified proteins of each strain in each xylem sap media. The table represents increased differentially abundant proteins in each strain under each xylem sap, which showed fold change 2 or more, and proteins which were identified only in each strain under each xylem sap media represented by at least 2 peptides.
- Figure 4 – missing information on statistical analyses, error bars, and n. This is a key result and should be explained more clearly in the results or legend.
>>> Statistical information has been added to figure legend.
- Section 3.3 in the results was particularly hard to follow at times. We implore the authors to write this section in a more digestible manner and/or create a figure on the findings.
>>> Text has been modified for more clarity.
- Page 6, line 249: fold changes were statistically significant at p<0.0. We presume the authors intended a p-value of 0.05. However, if this is not the case, the authors should clarify the cutoff of the value (i.e. 0.001 vs. 0.0001) and why a stringent approach is necessary.
>>> Corrected.
- Figure S1 is missing statistical analyses
>>> Added to the figure.
- Figure S2 should include quantification of ROS.
>>> Thank you for the valuable comment. As you know ROS is the first chemical barrier in plant defense. We stained the infection site to show the localized ROS accumulation upon infection. Non-inoculated plant leaves showed the slightly staining compared to infected ones. We believe that the histochemical assay that we provided here in the manuscript is suitable in providing an evidence of rapid localized ROS accumulation and its impact on disease progress, especially for the weakly aggressive strain DOAB232.
- In Table 2, it is unclear why some proteins identified are listed twice with each entry being present in one sap and then the other, whereas other proteins are listed once with presence in both saps (ex. CMN_00689 vs CMN_02051).
>>> Some of the identified proteins come with different accession numbers and different gene/locus. So, the same protein name may be repeated in the same table for that reason, for example Oxidoreductase.
- In Table 3, there are notable discrepancies between multiple gene loci identifiers and their protein annotations (ex. CMN_01917: Secreted lipase vs. Serine protease, family S1C). This may be due to formatting issues of the table but should be nevertheless checked and fixed as appropriate.
>>> The font size in the tables was changed to 8 instead of 9 tables were adjusted.
- For Figure 2, the clustering method should be listed (k-means vs. hierarchical cluster such as Ward’s method, etc.).
>>> The clustering method has been added to the legend.
Page 14, line 321: subtilase B (subtB) should be correct as sbtB; similarly, on page 16, line 401.
>>> This has been corrected.
- Page 15, line 353: It is unclear how long-term storage would have any impact on pathogen virulence. While the authors focused on proteins which were found wither only in Cn14-15-1 and/or in corn based xylem media, it is important to note that since many, if not most, of the plant targets are unknown.
>>> The authors hypothesized that long storage periods might have impact on the level of aggressiveness of Cn isolates. Dependent on the storage conditions, bacterial and fungal plant pathogens have been reported to lose virulence or even viability. Since Cn is generally not a potent pathogen, long-term storage ex-situ could lead to a slow but steady loss of traits required for interaction and disease induction in corn, especially for plasmid-borne traits. With respect to the next comment, the authors completely agree with reviewer 2 that this is a “black-box” in which plant targets of most of the Cn proteins are still unknown.
- Page 16, line 393: while the celluase activity referred to does show increased activity, it does not conclude that they are acting as virulence factors. Rather it shows that bacterial loci that act as virulence genes in other Clavibacter species are functional and have increased activity.
>>> Modified for more clarification.
- Page 16, line 406-407: It’s unclear how increased protease activity would lead the authors to conclude ‘…an accelerated action to deplete the native CXS proteins” when no targets of these proteins are known and casein is used as a substrate. This should be toned down.
>>> Sentence/paragraph has been re-phrased for clarification.
Round 2
Reviewer 1 Report
Prior comments have been addressed or revised satisfactorily.
Reviewer 2 Report
The authors have addressed the majority of my questions in the review. I think the MS is fine to publish.